# Fabrication of Metal/Carbon Nanotube Composites by Electro Chemical Deposition

Susumu Arai 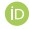

Department of Materials Chemistry, Faculty of Engineering, Shinshu University, Nagano 380-8553, Japan; araisun@shinshu-u.ac.jp

**Abstract:** Metal/carbon nanotube (CNT) composites are promising functional materials due to the various superior properties of CNTs in addition to the characteristics of metals, and consequently, many fabrication processes of these composites have been vigorously researched. In this paper, the fabrication process of metal/CNT composites by electrochemical deposition, including electrodeposition and electroless deposition, are comprehensively reviewed. A general introduction for fabrication of metal/CNT composites using the electrochemical deposition is carried out. The fabrication methods can be classified into three types: (1) composite plating by electrodeposition or electroless deposition, (2) metal coating on CNT by electroless deposition, and (3) electrodeposition using CNT templates, such as CNT sheets and CNT yarns. The performances of each type have been compared and explained especially from the view point of preparation methods. In the cases of (1) composite plating and (2) metal coating on CNTs, homogeneous dispersion of CNTs in electrochemical deposition baths is essential for the formation of metal/CNT composites with homogeneous distribution of CNTs, which leads to high performance composites. In the case of (3) electrodeposition using CNT templates, the electrodeposition of metals not only on the surfaces but also interior of the CNT templates is the key process to fabricate high performance metal/CNT composites.

**Keywords:** metal/carbon nanotube composite; electrochemical deposition; electrodeposition; electroless deposition; composite plating; carbon nanotube sheet; carbon nanotube yarn

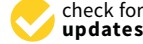



## 1. Introduction

Carbon nanotubes (CNTs) [1,2] have excellent mechanical characteristics such as high tensile strength and high elastic modulus, and also possess high thermal and electrical conductivity. Therefore, research into the practical applications of carbon nanotubes has been expanding into wide field, and composite materials of such nano-sized filler materials, such as polymer/CNT composites, have been studied expecting their innovative functions. Metal/CNT composites also have been investigated to enhance properties of metals and/or to give new innovative functions to metals. However, in general, the wettability of molten metals against CNTs is poor, resulting in difficulties of controlling the interface between the filler and matrix. In addition, since CNTs are nanosized fibrous materials and easily form aggregates, it is very difficult to form a metal/CNT composite with well-distributed CNTs in the metal matrix. To overcome these problems various methods, such as powder metallurgy, melting and solidification, and thermal spray, etc. have been challenged. The research in the fabrication methods of metal/CNT composites through different processes were reviewed by Bakshi et al. [3]. Electrochemical deposition is one of the most promising processes for fabricating metal/CNT composites with small size dimension. High-strength metal materials, high-thermal conductivity metal materials, high-abrasion resistance coatings, high-performance field emission elements, high-performance electrical contact materials, high-performance electromagnetic wave shielding materials, and so on, are specific applications.

Electrochemical deposition is roughly classified into electrodeposition and electroless deposition, and the fabrication processes of metal/CNT composites by the electrochemical deposition can be categorized into three types: (1) composite plating by electrodeposition or electroless deposition, (2) CNT coating by electroless deposition, and (3) electrodeposition using CNT templates (Figure 1). "Composite plating" is one of the electrochemical deposition techniques. CNTs are incorporated in deposited metal matrix during plating. In the case of "metal coating on CNTs by electroless deposition", the prepared metal-coated CNTs are mainly used as filler of composites, such as resin composites. In the case of "electrodeposition on CNT templates", CNT yarns or CNT sheets are used as CNT templates. The electrochemical deposition is a nano-scale or atomic scale process to fabricate metal materials and hence is effective to form atomic scall boundary between metals and CNTs. Moreover, this method is a wet process and consequently is likely advantageous to form metal/CNT composites with well-distributed CNTs in the metal matrix, especially in the case of the composite plating.

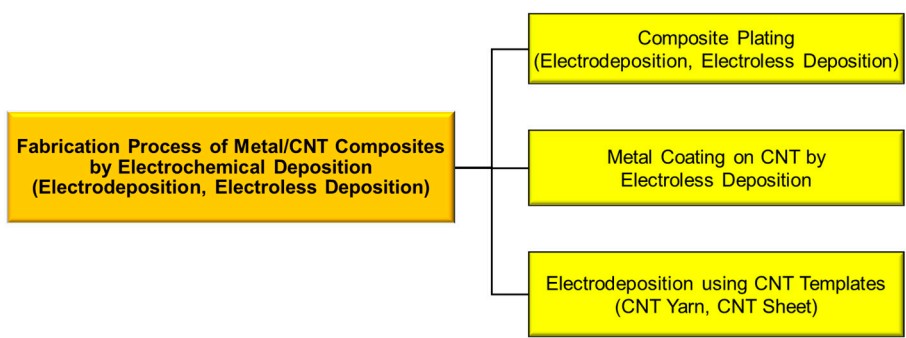

**Figure 1.** Classification of fabrication process for metal/CNT composites by electrochemical deposition.

Figure 2 shows the publication trend of peer-reviewed articles on the metal/CNT composites by electrochemical deposition. The articles counted are about the above three processes. The articles on metal-particle deposited CNTs by electroless deposition for the use of catalytic materials are not counted. Although there are fluctuations, total publication number tends to increase year by year (black circle). Looking at the breakdown, the articles on the composite plating are a lot (blue circle). The articles on "metal coating on CNTs by electroless deposition" have been published constantly even though the number is not so many (red circle). The number of articles on "electrodeposition using CNT templates" has been increasing since the first publication in 2010 (green circle).

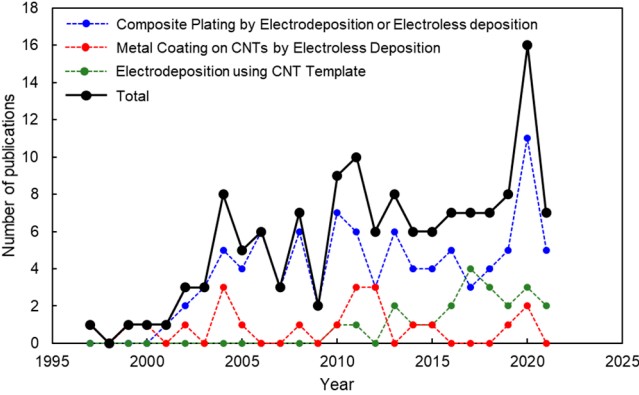

**Figure 2.** Publication number of articles on metal/CNT composites by electrochemical deposition during 1997–2021 (source: Web of Science).

In this review, based on the peer-reviewed articles, the research and development on the above three processes have been compared and explained especially from the viewpoint of preparation or fabrication methods. The purpose of this review is to show the research directions on the fabrication of metal/CNT composites by electrochemical deposition and to facilitate their practical applications.

## 2. Fabrication of Metal/CNT Composites Using Composite Plating by Electrodeposition or Electro Less Deposition

### 2.1. Composite Plating

Rough schematics of composite plating by electrodeposition and electroless deposition are displayed in Figures 3 and 4, respectively. In the case of electrodeposition, inert particles are dispersed homogeneously in a plating bath. When a voltage is applied, metal is electrodeposited on a cathode and the particles adsorb on the surface of the deposited metal. Then, the particles are embedded in depositing metal, resulting in a metal composite (Figure 3). In the case of CNT composite plating by electrodeposition, inert particles are dispersed homogeneously in a plating bath containing a reducing agent. When a substrate is soaked in the bath, metal is reductively deposited on the substrate accepting electrons from the reducing agent and, at the same time, the particles adsorb on the surface of the deposited metal. The particles are then embedded in depositing metal, resulting in a metal composite (Figure 4). In general, the substrate is pre-treated and catalyst particles, such as Pd particles, are fixed on the surface of the substrate before soaking into the plating bath. As far as was searched, the first article of the composite plating is on Cu/graphite composites by electrodeposition and was reported in 1928 [4]. Regarding the mechanism of the composite plating, several models have been proposed [5–9]. Guglielmi [5] proposed a mechanism based on two successive adsorption steps. In the first step, inert particles which are surrounded by adsorbed ions become loosely adsorbed on the cathode surface. In the second step, the particles lose the adsorbed ions and become strongly adsorbed on the cathode. Finally, the strongly adsorbed particles are embedded by the growing metal layer. Celis et al. [7] proposed a five-step model considering the effect of agitation and diffusion. In the first step, particles are surrounded by ions in the bulk of the solution. Then, in the second and third steps, the particles are transported by bath agitation to the hydrodynamic boundary layer and by diffusion through the diffusion layer, to the cathode. Finally, similar to Guglielmi's model, the particles are incorporated in two steps. However, there are still unclear parts, such as the effects on the mechanism of metal growth and the shape of inert particles. The history and details of the composite plating have been reviewed by Hovestad et al. [10].

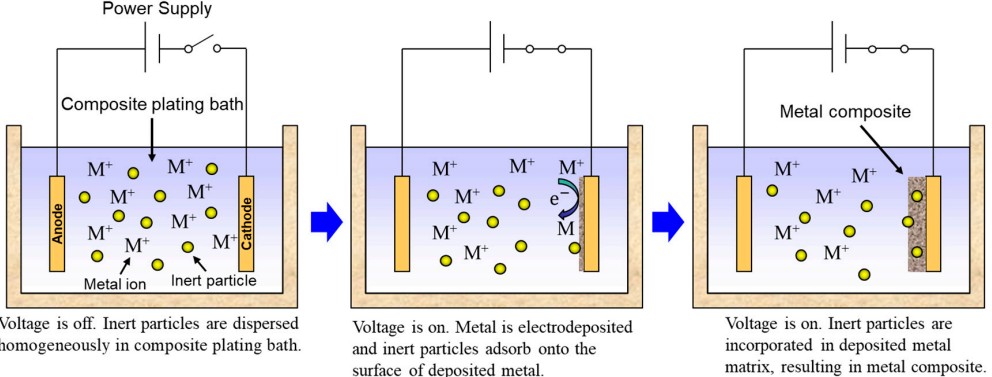

Voltage is off. Inert particles are dispersed homogeneously in composite plating bath.

Voltage is on. Metal is electrodeposited and inert particles adsorb onto the surface of deposited metal.

Voltage is on. Inert particles are incorporated in deposited metal matrix, resulting in metal composite.

**Figure 3.** Schematic of composite plating by electrodeposition.

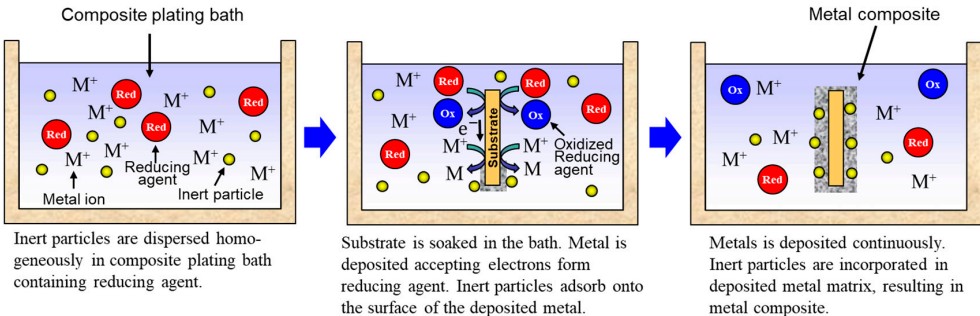

**Figure 4.** Schematic of composite plating by electroless deposition.

## 2.2. Preparation of Plating Bath for Metal/CNT Composite Plating

To fabricate metal/CNT composites with uniform distribution of CNTs, the preparation of plating baths with homogeneous dispersion of CNTs is important. In general, plating baths are aqueous solutions, while CNTs are hydrophobic. Therefore, hydrophilization of CNTs have been examined by the addition of surfactants or the direct introduction of hydrophilic groups on the surfaces of CNTs (Figure 5). The addition of surfactants in plating baths is a common method. Various kinds of surfactants [11–28], such as sodium dodecylbenzene sulfonate and sodium deoxycholate, have been examined for the homogeneous dispersion of CNTs in a pure water. However, effective surfactants for the dispersion in a pure water are not always effective in plating baths which contain great amounts of ions. Moreover, even if the surfactant is effective for the dispersion of CNTs in a plating bath, CNTs are not always co-deposited by electrochemical deposition. Therefore, the selection of appropriate surfactants is essential. Since the surfactants are likely incorporated in deposited metal matrix during electrochemical deposition, the concentration of surfactants should be examined. On the contrary, the direct introduction of hydrophilic groups, such as -COOH, onto the surfaces of CNTs has been examined using a chemical treatment [29], a plasma treatment [30], a heat treatment [31], and so on. These methods destroy the sp$^2$ carbon bonding of the surfaces of CNTs. Therefore, the conditions of the treatments should be examined.

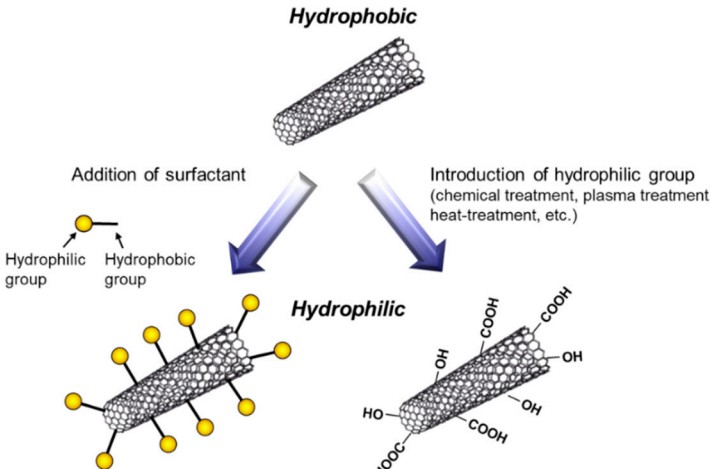

**Figure 5.** Schematic of hydrophilization of CNTs.

On the contrary, CNTs are nanosized fibrous material and consequently tend to aggregate. In particular, SWCNTs have the thinnest (ca. 1–4 nm in diameter) among the various types of CNTs and can thus easily form aggregates referred to as bundles (Figure 6). To fabricate metal/SWCNT composites with a homogeneous distribution of SWCNTs as the primary particles in the metal matrix, the preparation of composite plating baths that contain well-disintegrated and well-dispersed SWCNTs is necessary.

For the preparation of metal/SWCNT composite plating baths, application of mechanical (or physical) disintegrations is effective in addition to adding dispersing agents and/or introduction of hydrophilic groups. Yang et al. [32] used an ultrasonification in addition to both a heat treatment and an addition of surfactants to prepare an Ni-P alloy/SWCNT composite plating bath for electroless deposition. A Cu/SWCNT composite plating bath for electroless deposition using a collision type mechanical disintegrator in addition to adding surfactants has also been prepared and Cu/SWCMT composite films with homogenously distributed SWCNTs have been obtained from the bath [33].

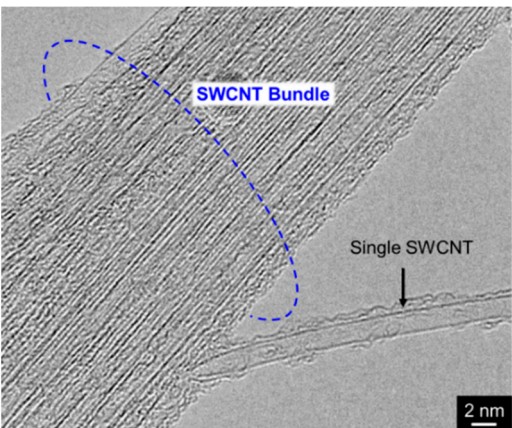

**Figure 6.** TEM image of SWCNT bundle.

### 2.3. Unique Feature of Composite Plating Using CNTs as Inert Particles

Since a single CNT, especially multi-walled CNT (MWCNT) has a fibrous shape with large aspect ratio in addition to a high electrical conductivity in the axis direction. Therefore, composite plating using CNTs as inert particles often shows a unique feature unlike other composite plating using insulation particles such as $Al_2O_3$ particles. The schematic of the unique feature is showed in Figure 7 [34]. When a part of a MWCNT is incorporated in the deposited metal matrix during electrodeposition, the metal can be electrodeposited not only on the deposited metal but also on the protruding edge (a defect site) of the MWCNT. If the defect sites exist on the sidewall of the MWCNT, the metal can also be electrodeposited on the defect sites.

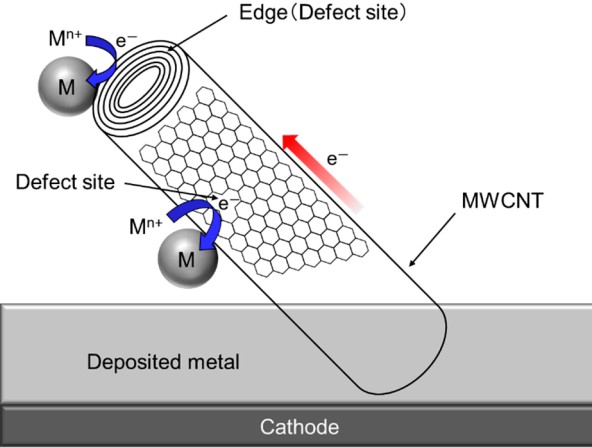

**Figure 7.** Schematic of unique phenomenon of composite plating using MWCNTs as inert particles. (Figure 7 is adapted from reference [34]).

Using this unique phenomena, powder Cu/MWCNT composites could be obtained [35]. Figure 8a displays the surface morphology of Cu/MWCNT composites just after the electrodeposition. Many Cu/MWCNT composites particles are seen. These particles are fixed

loosely on the cathode substrate and can be separated easily by ultrasonification. Figure 8b displays the morphology of the Cu/MWCNT composite powder after the separation from the substrate by ultrasonification. A large number of MWCNTs stick out from the Cu particles, resulting in a sea urchin shape. The size of the Cu spheres is 2–15 μm.

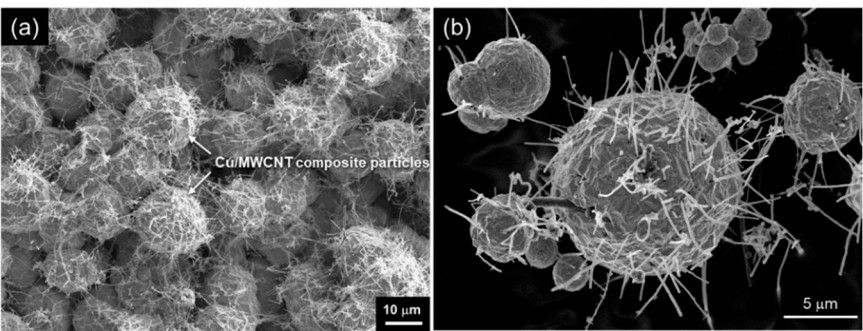

**Figure 8.** SEM images of (**a**) Cu/MWCNT composite immediately after electrodeposition and (**b**) Cu/MWCNT composite powder separated by ultrasonification. (Figure 8 is adapted from reference [35]).

### *2.4. Fabrication of Metal/CNT Composites Using Composite Plating by Electrodeposition*

Regarding the number of published articles on metal/CNT composite plating using electrodeposition, those on Ni/CNT and Cu/CNT composite plating are large. In the following section, the summaries of articles on various metal/CNT composites using electrodeposition are described. Fabrication conditions in these articles are listed in Table 1.

### 2.4.1. Ni/CNT and Ni Alloy/CNT Composites

Chen et al. [36,37] have published the paper on Ni/CNT composites by electrodeposition. They used a Watts type bath as the base bath and employed MWCNTs prepared by themselves. The composite plating was conducted under ultrasonifacation to disperse MWCNTs and the composite films obtained showed improved tribological properties. They also used a cationic surfactant of cetyltrimethyl ammonium bromide for MWCNTs and the resultant Ni/MWCNT composite films showed improved corrosion resistance than a Ni film [38]. A polyacrylic acid was one of the effective dispersants for large-sized MWCNTs in acidic Ni/MWCNT composite plating baths [34,39–41]. Pulse-reverse electrodeposition processes were effective to control morphologies and CNT contents of Ni/CNT composite films [42]. Thermal conductivity of Ni/MWCNT composite films was evaluated and a Ni/0.7 mass% MWCNT composite film showed a maximum value of 109 W m$^{-1}$ K$^{-1}$, which is about twice that of a Ni film without MWCNT [39]. The internal stresses of Ni/MWCNT composite films were studied and a composite film with low internal stress could be fabricated by controlling concentrations of additives [40]. Guo et al. studied the effects of pulse-reverse parameter [43], current density [44] and surfactants [45] on the properties of the composite films and demonstrated the increases of hardness, the increases of corrosion resistance, and the decrease in mechanical properties of the composite films, respectively. Dai et al. [46] prepared Ni/MWCNT composite films from a bright Watts bath and demonstrated higher tensile strength of the composite films than Ni films. Jeon et al. [47] fabricated Ni/CNT composite films using a mixed dispersant of sodium lauryl sulfate and hydroxy propyl cellulose for MWCNTs and showed the relationship between CNT content and fracture stress. Since CNTs have a solid lubricity [48,49], Ni/CNT composite films showed lower coefficient of friction even under dry conditions (without liquid lubricants) [41]. Electrodeposition of Ni/CNT composite films from non-aqueous solvents were also studied to achieve improved dispersion of CNTs. Martis et al. [50] employed a choline chloride/urea based deep eutectic solvent and demonstrated excellent dispersion stability of oxidized MWCNTs in the plating bath. The Ni/MWCNT composite films prepared using the bath showed improved corrosion resistance. Liu et al. [51] electrodeposited Ni/MWCNT composite films from a plating bath using choline chloride/carbamide deep

eutectic solvent and demonstrated their higher hardness, higher elastic modulus, and lower co-efficient of friction compared with a Ni film. Kim et al. [52] studied the hardness and corrosion characteristics of the Ni/MWCNT composite films using a plating bath containing a mixed dispersant of sodium lauryl sulfate and hydroxy propyl cellulose [47]. Yang [53] proposed a cyclic voltammetric process to control the morphologies of Ni/MWCNT composite films using a plating bath containing polyvinylpyrrolidone as the dispersing agent. Suzuki et al. [54] demonstrated that Ni/MWCNT composite coating was effective on the improvement in tool life of electroplated diamond tools. Wang et al. [55] prepared MWCNTs wrapped by a polydopamine and demonstrated that the Ni/polydopamine-modified MWCNT composite films showed excellent tribological properties. Prasannakumar et al. [56] evaluated corrosion protection persistence of Ni/MWCNT composite coating on steel. Jyotheender et al. [57] studied the corrosion behavior of Ni/MWCNT composite coatings from the viewpoint of the grain boundary.

Several articles on Ni alloy/CNT composites have also been reported. Shi et al. [58] studied corrosion properties of Ni-P alloy/MWCNT coatings. Shi et al. [59] investigated mechanical properties of Ni-Co alloy/MWCNT composite coatings. Suzuki et al. [60] investigated tribological properties of Ni-P alloy/MWCNT composite films. Arora et al. [61] studied corrosion behavior of Ni-Co alloy/MWCNT composite coatings.

### 2.4.2. Cu/CNT Composites

Polyacrylic acid was an effective dispersing agent for large-sized MWCNTs in acidic copper sulfate baths [35,62–72]. Cu/MWCNT composite powder [35] and films [62] were fabricated by electrodeposition using a sulfate bath containing a polyacrylic acid as the dispersing agent. Cu/cup-stacked CNT composite films also could be fabricated [63]. Yang et al. [73] fabricated Cu/SWCNT composite films under an ultrasonic condition and evaluated electrical resistivity and microhardness of the composites. Chai [74] et al. formed Cu/SWCNT composite films and evaluated their mechanical properties. Due to their high aspect ratio, chemical inertness, and electric conductivity, CNTs have been watched with keen interest as potential field emission electron sources [75]. Patterned MWCNT emitters were fabricated using photolithography and composite plating [64] (Figure 9). This technology can be expected to be applied to field emission displays. The effects of various electrodeposition conditions on the microstructures of Cu/MWCNT composite films were studied, and the internal stress, hardness, and electrical conductivity of the composite films were evaluated [65]. A pulse-reverse electrodeposition was effective for the increase in MWCNTs in Cu/MWCNT composite films [66]. Qin et al. [76] studied the electrochemical reduction behavior of Cu/MWCNT composite plating. A mechanism for co-deposition of MWCNTs with Cu was studied [77] in relation to Guglielmi's model [5]. Cu foils have been used as current collectors of lithium-ion battery anodes. The volume of next generation active materials of the lithium-ion battery (LIB) anodes, such as Sn, Si, and SiO, are greatly changed during the charge–discharge processes. Therefore, the adhesion strength between the active materials layer and copper current collectors is important. Uneven Cu/MWCNT composite films on which MWCNTs protrude were applied as the current collectors of Sn-based [78] and Si-based [79] LIB anodes and performed excellent charge–discharge cycle properties. Feng. et al. [67] fabricated Cu/MWCNT composite films by a periodic pulse reverse electrodeposition using nano diamond particles as the dispersing agent for MWCNTs. Zheng et al. [68] fabricated bulk Cu/MWCNT composites by sintering Cu/MWCNT composite powder produced by composite plating. Wang et al. [69,70] fabricated bulk Cu/MWCNT composites in a similar way and evaluate electrical and mechanical properties. Chen et al. [80] and Wang et al. [81] fabricated bulk Cu/MWCNT composite by a combination of composite plating and spark plasma sintering and evaluated their mechanical properties. Fu et al. [82] showed the effects of heat treatments on the electrical conductivity of Cu/MWCNT composite films. Ning et al. [71] fabricated Cu/MWCNT composite films by a jet electrodeposition and evaluated the tribological properties of them. Raja et al. [83] formed Cu/SWCNT composite films

neither pre-treatment of CNTs nor addition of surfactants and studied their microstructure. Multi-layered copper foils reinforced by a Cu/SWCNT composite film were fabricated and their tensile strengths were evaluated [72]. Li et al. [84] fabricated Cu/MWCNT composite films by a pulse-reverse electrodeposition and evaluated their electrical and mechanical properties. Aliyu et al. [85] studied the relationship between growth texture, crystalline size, lattice strain and corrosion behavior of Cu/MWCNT composite films and showed the optimum condition to form the composite films.

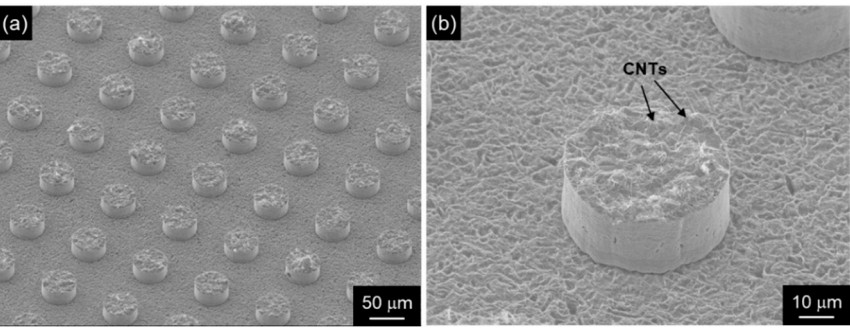

**Figure 9.** SEM images of patterned CNT emitters fabricated by composite plating assisted with photolithography: (**a**) low magnification, (**b**) high magnification. (Figure 9 is adapted from reference [73]).

### 2.4.3. Zn/CNT and Zn Alloy/CNT Composites

Most of articles on Zn/CNT composite films were investigated to improve corrosion properties. Praveen et al. [86] fabricated Zn/MWCNT composite films from an acid sulphate bath. They showed higher corrosion resistance of the Zn/MWCNT composite films than Zn films. Tseluikin et al. [87] fabricated Zn/MWCNT composite films from an alkaline bath and evaluated their tribological properties and corrosion resistance. They also fabricated Zn-Ni alloy/MWCNT composite films in a pulse-reverse current mode and evaluated their tribological and corrosion properties [88]. Jyotheender et al. [89] studied the correlation between grain the boundary character and the corrosion behavior of Zn/MWCNT composite films.

### 2.4.4. Cr/CNT Composites

Most of papers on Cr/CNT composite films were studied to improve mechanical properties. In addition, most plating baths used were environmentally friendly trivalent Cr baths. Liu et at. [90] fabricated Cr/MWCNT composite films from a trivalent Cr bath and evaluated their mechanical properties. Shukla et al. [91] and Tripathi et al. [92] demonstrated superior mechanical properties of Cr/MWCNT composite films and Cr/(yttria-stabilized- zirconia + MWCNT) composite films.

### 2.4.5. Co/CNT and Co Alloy/CNT Composites

Co and its alloy or composite films have high hardness, good wear and corrosion resistance and, therefore, they have been expected to replace Cr plating films using hexavalent Cr baths [93]. Su et al. [94] fabricated Co/MWCNT composite films from a sulfate bath and demonstrated their superior tribological and corrosion properties of the Co/MWCNT composite films than Co films. Co/MWCNT composite films with different-sized MWCNTs were fabricated and their tribological properties at high temperatures were studied [95]. Field emission properties of Co/MWCNT composite films were also investigated [96]. Pereira et al. [97] fabricated Co/MWCNT composite films from a choline-chloride-base deep eutectic solvent and studied their electrical and corrosion properties. Co-W alloy/MWCNT composite films were fabricated and their tribological properties at high temperatures were investigated [98]. Anand et al. [99] prepared Co-W alloy/MWCNT composite films by a pulse electrodeposition and evaluated their tribological and corrosion properties.

### 2.4.6. Au/CNT and Ag/CNT Composites

Au plating and Ag plating have been widely used for electrical contact parts. Fugishige et al. [100] fabricated Au/MWCNT composite films from a non-cyanide sulfide bath and studied their hardness and tribological properties. Fujishige et al. [101] also studied electrical contact characteristics of Ag/MWCNT composite films corroded using $H_2S$ gas and showed their superior electrical contact characteristics against sulfidation. These films were prepared from a cyanide bath. According to strong demands for non-cyanide Ag plating technology, the developments of non-cyanide baths have been advanced. Recently, as the non-cyanide bath for the electrodeposition of Ag/CNT composite films, an iodide bath was developed [102]. The Ag/MWCNT composite films fabricated from the iodide bath showed same electrical conductivity as a pure Ag film and lower coefficient of friction than a pure Ag film. In addition, this composite film showed sulfidation resistance due to hydrophobic nature of MWCNTs [103] (Figure 10). Brandao et al. [104] fabricated CNT/Ag composites through a pulse reverse electrodeposition using a deep eutectic solvent.

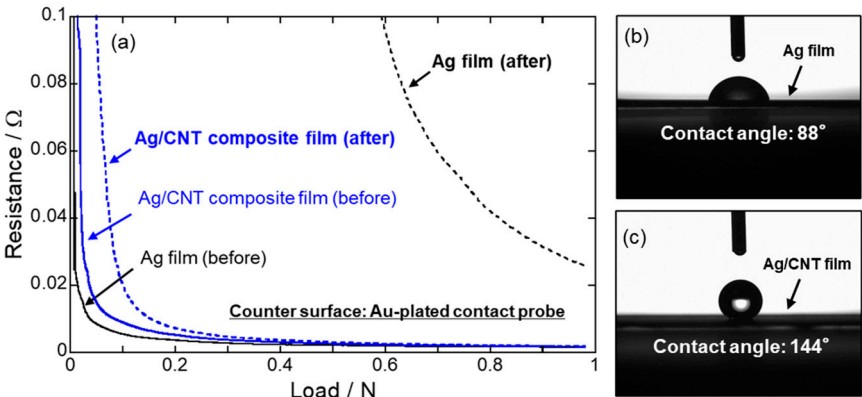

**Figure 10.** (**a**) Electrical contact resistance of Ag and Ag/CNT composite films before and after a sulfidation test. (**b**) and (**c**) are images of sessile water droplets on Ag film and Ag/MWCNT composite film, respectively. (Figure 10 is adapted from [103], Gold open access: CC BY-NC-ND 4.0, 2021).

### 2.4.7. Al/CNT Composites

Al could not be electrodeposited from aqueous solutions due to the narrow potential window (1.23 V). Therefore, Al/CNT composite films are electrodeposited from plating baths using non-aqueous solvents with a wide potential window. Yatsushiro et al. [105] electrodeposited Al/MWCNT composite films from an $AlCl_3$-1-ethyl-3-methylimidazolium chloride molten salt. Zhang et al. [106] fabricated Al/MWCNT composite films from a plating bath using a diglyme as the solvent and showed their improved hardness compared with a pure Al film.

### 2.4.8. Other Metal/CNT Composites

Hu et al. [107] prepared Pb-Sn composite films from a fluoroborate bath and studied corrosion properties of them. Brandao et al. [108] fabricated Sn/MWCNT composites using a choline chloride-based ionic liquid.

### 2.5. Fabrication of Metal/CNT Composites Using Composite Plating by Electroless Deposition

Regarding the number of published articles on metal/CNT composite plating using electroless deposition, those on the Ni-P alloy/CNT is large. In the case of electroless deposition of Ni, phosphorous compounds such as sodium hypophosphite ($NaH_2PO_2$) are usually used as the reducing agent and the P derived from the $NaH_2PO_2$ is co-deposited with Ni, resulting in Ni-P alloy deposit. Most of the purpose of the fabrication of Ni-P alloy/CNT composites is the improvement of tribological properties. Several articles on Cu/CNT composite plating have been reported. In the following section, the summaries of articles on metal/CNT composites by electroless deposition are described. Fabrication conditions in these articles are listed in Table 2.

Table 1. Fabrication conditions of metal/CNT composites using composite plating by electrodeposition.

| Metal | CNT | Treatment of CNT | Base Plating Bath | Surfactant | Remarks | Year | Ref. |
|-------|-----|------------------|-------------------|------------|---------|------|------|
| Ni | MWCNT | Chemical treatment | Dull Watts bath | Sodium lauryl sulfate | Corrosion behavior | 2020 | [57] |
| Ni | MWCNT | Chemical treatment | Dull Watts bath | Sodium lauryl sulfate | Corrosion protection | 2020 | [56] |
| Ni | MWCNT | Wrapped by polydopamine | Dull Watts bath | Non | Wear and corrosion resistance | 2019 | [55] |
| Ni | MWCNT | Non | Ionic liquid (choline chloride/carbamide) | Non | Non-aqueous solvent | 2017 | [51] |
| Ni | MWCNT | Non | Sulfamate bath | Cationic surfactant, compound name is unknown | Improvement in tool life | 2014 | [54] |
| Ni | MWCNT | Non | $NiSO_4$+NaCl | Polyvinylpyrrolidone | Cyclic voltametric route | 2011 | [53] |
| Ni | MWCNT | Ball milling | Bright Watts bath | Sodium lauryl sulfate and Hydroxypropylcellulose | Corrosion behavior | 2011 | [52] |
| Ni | MWCNT | Chemical treatment | Choline chloride/urea | Non | Non-aqueous solvent | 2010 | [50] |
| Ni | MWCNT | Non | Bright Watts bath | Polyacrylic acid | Solid lubrication | 2008 | [41] |
| Ni | MWCNT | Ball milling | Watts type bath | Sodium lauryl sulfate, Cetyltrimethylammonium bromide | Effects of surfactants | 2008 | [45] |
| Ni | MWCNT | Chemical treatment | Dull Watts bath | Non | Effects of current density | 2008 | [44] |
| Ni | MWCNT | Ball milling | Bright Watts bath | Sodium lauryl sulfate and Hydroxypropylcellulose | Mechanical properties | 2008 | [47] |
| Ni | MWCNT | Non | Bright Watts bath | Non | Mechanical properties | 2008 | [46] |
| Ni | MWCNT | Non | Bright Sulfamate bath | Polyacrylic acid | Low internal stress | 2007 | [40] |
| Ni | MWCNT | Non | Dull Watts bath | Non | Pulse-reverse parameter | 2007 | [43] |
| Ni | MWCNT | Non | Bright Watts bath | Polyacrylic acid | Thermal conductivity | 2006 | [39] |
| Ni | MWCNT | Non | Dull Watts bath | Poly(diallyldimethylammonium chrolide) | Pulse-reverse electrodeposition | 2005 | [42] |
| Ni | MWCNT | Chemical treatment | Dull Watts bath | Cetyltrimethylammonium bromide | Corrosion behavior | 2005 | [38] |
| Ni | MWCNT | Non | Dull Watts bath | Polyacrylic acid | Ni deposition on incorporated CNT | 2004 | [34] |
| Ni | MWCNT | Ball milling | Dull Watts bath | Non | CNT content | 2002 | [37] |
| Ni | MWCNT | Ball milling | Dull Watts bath | Non | Tribological property | 2001 | [36] |
| Ni-Co | MWCNT | Chemical treatment | Dull Watts bath + Co salt | Non | Corrosion behavior | 2019 | [61] |
| Ni-P | MWCNT | Non | Dull Watts bath + citric acid + P compound | Polyacrylic acid | Tribological properties | 2010 | [60] |

**Table 1.** *Cont.*

| Metal | CNT | Treatment of CNT | Base Plating Bath | Surfactant | Remarks | Year | Ref. |
|---|---|---|---|---|---|---|---|
| Ni-Co | MWCNT | Non | Dull Watts bath + Co salt | Compound name is unknown | Mechanical and tribological properties | 2006 | [59] |
| Ni-P | MWCNT | Non | Ni salts + citric acid + P compounds | Compound name is unknown | Corrosion properties | 2004 | [58] |
| Cu | MWCNT | Chemical treatment | Citric bath | Non | Corrosion behavior | 2021 | [85] |
| Cu | MWCNT | Chemical treatment | Sulfate bath | Non | Pulse reverse, electrical conductivity | 2020 | [84] |
| Cu | MWCNT | Chemical treatment? | Sulfate bath | Non-ionic surfactants, Compound name is unknown | Mechanical properties, Microlaminated structure | 2020 | [81] |
| Cu | SWCNT | Non | Sulfate bath | Stearyltrimethylammonium chloride | Mechanical properties | 2020 | [72] |
| Cu | SWCNT | Non | Sulfate bath | Non | Microstructure | 2019 | [83] |
| Cu | MWCNT | Non | Sulfate bath | Sodium lauryl sulfate | Jet electrodeposition, Tribological properties | 2019 | [71] |
| Cu | MWCNT | Non | Sulfate bath | Polyacrylic acid | Current collector for LIB anode | 2019 | [79] |
| Cu | MWCNT | Chemical treatment | Sulfate bath | Stearyltrimethylammonium bromide | Electrical conductivity, Corrosion resistance | 2018 | [82] |
| Cu | MWCNT | Non | Sulfate bath | Non-ionic surfactants, Compound name is unknown | Mechanical properties, Laminated structure | 2018 | [80] |
| Cu | MWCNT | Chemical treatment | Sulfate bath | Non | Cu/CNT powder + powder metallurgy | 2018 | [70] |
| Cu | MWCNT | Chemical treatment | Sulfate bath | Non | Cu/CNT powder + powder metallurgy | 2018 | [69] |
| Cu | MWCNT | Chemical treatment | Sulfate bath | Non | Cu/CNT powder + powder metallurgy | 2017 | [68] |
| Cu | MWCNT | Chemical treatment | Commercially available | Nano diamond | Periodic pulse reverse electrodeposition | 2016 | [67] |
| Cu | MWCNT | Non | Sulfate bath | Polyacrylic acid | Current collector for LIB anode | 2016 | [78] |
| Cu | MWCNT | Non | Sulfate bath | Polyacrylic acid | Co-deposition mechanism of CNT | 2013 | [77] |
| Cu | MWCNT | Non | Sulfate bath | Non | Electrochemical reduction behavior | 2011 | [76] |
| Cu | MWCNT | Non | Sulfate bath | Polyacrylic acid | Pulse-reverse | 2011 | [66] |
| Cu | MWCNT | Non | Sulfate bath | Polyacrylic acid | Surface morphology, Hardness, Internal stress | 2010 | [65] |
| Cu | MWCNT | Non | Sulfate bath | Polyacrylic acid | Patterned field emitter | 2008 | [64] |
| Cu | SWCNT | Non | Sulfate bath | Commercial products | Mechanical properties | 2008 | [74] |
| Cu | SWCNT | Chemical treatment | Sulfate bath | Cetyltrimethylammonium chloride | Mechanical properties | 2008 | [73] |
| Cu | Cup-stacked CNT | Non | Sulfate bath | Polyacrylic acid | Various CNTs | 2005 | [63] |
| Cu | MWCNT | Non | Sulfate bath | Polyacrylic acid | Microstructure | 2004 | [62] |

**Table 1.** *Cont.*

| Metal | CNT | Treatment of CNT | Base Plating Bath | Surfactant | Remarks | Year | Ref. |
|---|---|---|---|---|---|---|---|
| Cu | MWCNT | Non | Sulfate bath | Polyacrylic acid | Cu/MWCNT composite powder | 2003 | [35] |
| Zn | MWCNT | Chemical treatment | Sulfate bath | Cetyltrimethylammonium bromide | Corrosion resistance | 2021 | [89] |
| Zn | MWCNT | Non | Zincate bath | Unknown | Pulse electrodeposition, Corrosion resistance | 2020 | [87] |
| Zn | MWCNT | Chemical treatment | Sulfate bath | Cetyltrimethylammonium bromide | Corrosion resistance | 2007 | [86] |
| Zn-Ni | MWCNT | Non | Chloride bath | Non | Pulse reverse, Tribological and Corrosion properties | 2016 | [88] |
| Cr | MWCNT | Non | Trivalent Cr bath | Sodium lauryl sulfate | Tribological properties, Corrosion resistance | 2020 | [92] |
| Cr | MWCNT | Non | Trivalent Cr bath | Sodium lauryl sulfate | Tribological properties | 2018 | [91] |
| Cr | MWCNT | Non | Trivalent Cr bath | Non | Mechanical properties | 2009 | [90] |
| Co | MWCNT | Non | Choline chloride/urea | Non | Non-aqueous solvent | 2017 | [97] |
| Co | MWCNT | Non | Sulfate bath | Polyacrylic acid | Field emission properties | 2013 | [96] |
| Co | MWCNT | Non | Sulfate bath | Polyacrylic acid | Tribological properties | 2013 | [95] |
| Co | MWCNT | Acid-treatment | Sulfate bath + citrate | Sodium lauryl sulfate | Tribological properties, Corrosion properties | 2013 | [94] |
| Co-W | MWCNT | Non | Co salt + Tungstate + Citrate | Polyacrylic acid | Tribological properties Corrosion properties | 2015 | [99] |
| Co-W | MWCNT | Non | Co salt + Tungstate + Citrate | Polyacrylic acid | Tribological properties | 2013 | [98] |
| Au | MWCNT | Non | Sulfite bath | Stearyltrimethylammonium chloride | Electrical conductivity, Tribological properties | 2009 | [100] |
| Ag | MWCNT | Non | Choline chloride + glycerol | Poly (N-vinyl pyrrolidone) | Pulse reverse electrodeposition | 2021 | [104] |
| Ag | MWCNT | Non | Iodide bath | Non | Electrical contact resistance against $H_2S$ gas | 2021 | [103] |
| Ag | MWCNT | Non | Iodide bath | Non | Hardness, Electrical and Tribological properties | 2020 | [102] |
| Ag | MWCNT | Non | Cyanide bath | Unknown | Electrical contact resistance against $H_2S$ gas | 2010 | [101] |
| Al | MWCNT | Acid treatment | Diethylene glycol dimethyl ether | Non | Hardness | 2020 | [106] |
| Al | MWCNT | Non | 1-ethyl-3-methylimidazolium chloride | Non | Hardness | 2006 | [105] |
| Sn | MWCNT | Non | Choline chloride + ethylene glycole | Non | Nucleation study | 2019 | [108] |
| Pb-Sn | MWCNT | Acid treatment | Fluoroborate bath | Polyacrylic acid | Corrosion resistance | 2010 | [107] |

### 2.5.1. Ni-P Alloy/CNT Composites

Chen et al. [109–111] fabricated Ni-P alloy/MWCNT composite coatings by electroless deposition and demonstrated their superior tribological properties of the composite coatings. They used a cationic surfactant of cetyltrimethylammonium bromide in addition to the ball-milling of MWCNTs to disperse the MWCNTs homogeneously in the plating bath. Yang et al. [112] prepared Ni-P alloy/SWCNT composite films and studied their tribological and corrosion properties. Chen et al. [113] prepared Ni-P alloy/MWCNT composite films and examined their tribological properties under dry conditions. Li et al. [114] prepared short MWCNTs without chemical treatment or heat treatment which would damage the CNTs. They evaluated the tribological properties of the prepared Ni-P alloy/MWCNT composite coatings. Electroless Ni-P alloy plating has been used for printed circuit board pads. Gu et al. [115] studied shearing behavior between Ni-P alloy/MWCNT composite film and a solder after soldering. Industrially, Ni-P alloy plating films are roughly classified into three types (low P type, medium P type, and high P type) according to the difference in composition. The low, medium, and high P types contain 2–4 mass% P, 7–9 mass% P, and 12–13 mass% P, respectively. Park et al. [116] showed that a high-speed agitation by a mixing blade has efficiently cut large-diameter MWCNTs without serious damages. Electroless plating has the significant advantage that it can be used to fabricate films not only on conductive materials, such as metals, but also on insulators, such as resins and ceramics. Various types of Ni-P alloy/MWCNT composite films were fabricated on an ABS resin substrate and their adhesion strength and tribological properties were evaluated [117] (Figure 11). Firrzbakht et al. [118] prepared electroless coated Mg power. Zhao et al. [119] applied a mechanical attrition-assisted electroless deposition. Alishahi et al. [120] evaluated corrosion and tribological properties of Ni-P alloy/MWCNT composite coatings. Meng et al. [121] studied the milling effects on MWCNTs using wet-milling in comparison with ordinary dry-milling on the fabrication and tribological properties of Ni-P alloy/MWCNT composite coatings. Lopes de Oliveira et al. [122] researched corrosion behavior of Ni-P alloy/MWCNT composite coatings on a pipeline steel.

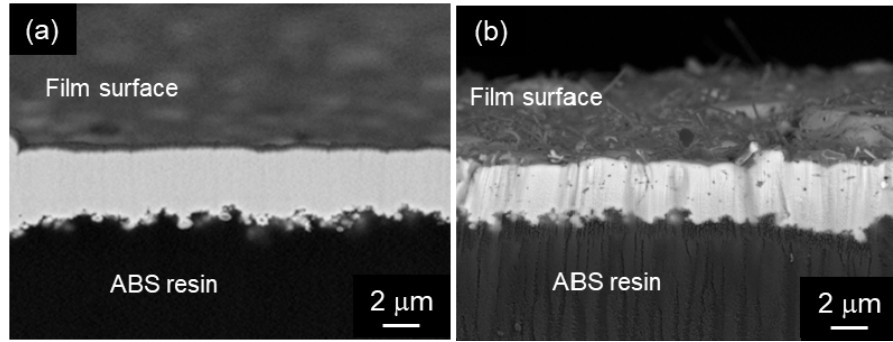

**Figure 11.** Cross-sectional SEM images of (**a**) electroless Ni-P alloy and (**b**) electroless Ni-P alloy/MWCNT composite film on an ABS resin substrate. (Figure 11 is adapted from [117]).

### 2.5.2. Cu/CNT Composites

Formaldehyde has been employed as a reducing agent for electroless Cu plating. However, according to the harmful to the human body, other reducing agents have been examined. Cu/MWCNT composite films with various sized MWCNTs were fabricated using glyoxylic acid as the reducing agent and their tribological properties were studied [123]. Cu/SWCNT composite films with homogeneously distributed SWCNTs were also fabricated using a mechanical disintegrator for the SWCNTs disintegration in the plating bath [33] (Figure 12).

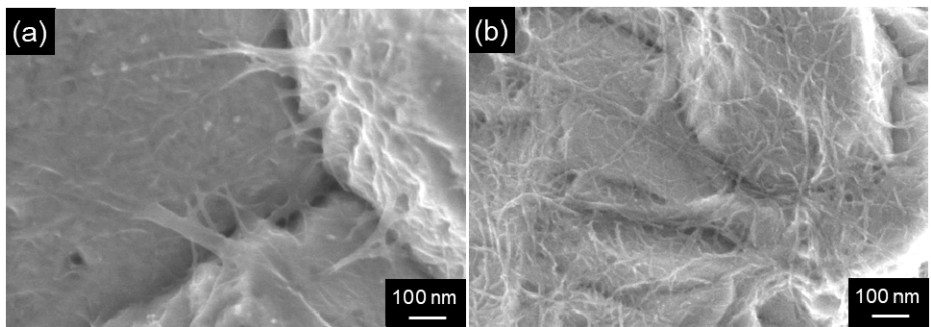

**Figure 12.** Surface SEM images of Cu/SWCNT composite films by electroless deposition formed from the plating baths treated by different disintegration methods. (**a**) an ultrasonic homogenizer, (**b**) a collision mechanical atomizer. (Figure 12 is adapted from [33]).

### 2.5.3. Other Metal/Composites

Goel et al. [124] prepared Co-P alloy/MWCNT composite coating on a grain-oriented electrical steel to enhance magnetic properties.

**Table 2.** Fabrication conditions of metal/CNT composites by electroless deposition.

| Metal | CNT | Pre-Treatment of CNT | Reducing Agent | Surfactant | Remarks | Year | Ref. |
|-------|-----|----------------------|----------------|------------|---------|------|------|
| Ni-P | MWCNT | Non | $NaH_2PO_2$ | Sodium lauryl sulfate | Tribological properties, Corrosion resistance | 2021 | [122] |
| Ni-P | MWCNT | Ball milling | $NaH_2PO_2$ | Cetyltrimethylammonium bromide | Tribological properties | 2012 | [121] |
| Ni-P | MWCNT | Ball milling, Chemical treatment | $NaH_2PO_2$ | Commercial product | Tribological properties, Corrosion resistance | 2012 | [120] |
| Ni-P | MWCNT | Chemical treatment Ball milling | $NaH_2PO_2$ | Sodium lauryl sulfate | Mechanical attrition, Tribological properties | 2012 | [119] |
| Ni-P | MWCNT | $HNO_3$ | Commercial product | Commercial product | Substrate: Mg powder | 2011 | [118] |
| Ni-P | MWCNT | Non | $NaH_2PO_2$ | Stearyltrimethylammonium chloride | Substrate: ABS resin Tribological properties | 2011 | [117] |
| Ni-P | MWCNT | Non | $NaH_2PO_2$ | Stearyltrimethylammonium chloride | Various P content, Tribological properties | 2010 | [116] |
| Ni-P | MWCNT | Chemical treatment | $NaH_2PO_2$ | Unknown | Effects on solder joint | 2009 | [115] |
| Ni-P | MWCNT | Chemical treatment | $NaH_2PO_2$ | Cetyltrimethylammonium bromide | Tribological properties | 2009 | [114] |
| Ni-P | MWCNT | Chemical treatment | $NaH_2PO_2$ | unknown | Tribological properties | 2006 | [113] |
| Ni-P | MWCNT | Ball milling | $NaH_2PO_2$ | Compound name is unknown | Hardness, Corrosion resistance | 2005 | [112] |
| Ni-P | SWCNT | Heat treatment | $NaH_2PO_2$ | Compound name is unknown | Tribological properties | 2004 | [32] |
| Ni-P | MWCNT | Ball milling | $NaH_2PO_2$ | Cetyltrimethylammonium bromide | Tribological properties | 2003 | [111] |
| Ni-P | MWCNT | Ball milling | $NaH_2PO_2$ | Cetyltrimethylammonium bromide | Tribological properties | 2003 | [110] |
| Ni-P | MWCNT | Ball milling | $NaH_2PO_2$ | Cetyltrimethylammonium bromide | Tribological properties | 2002 | [109] |
| Cu | SWCNT | Non | CHOCOOH | Sodium lauryl sulfate Hydroxypropylcellulose | Mechanical disintegration, | 2016 | [33] |
| Cu | MWCNT | Non | CHOCOOH | Sodium lauryl sulfate Hydroxypropylcellulose | Various CNTs Tribological properties | 2014 | [123] |
| Co-P | MWCNT | Non | $NaH_2PO_2$ | Non | Magnetic properties | 2016 | [124] |

### 3. Metal-Coated CNTs by Electroless Deposition

#### 3.1. Fabrication Process

A fabrication process of metal-coated CNTs by an autocatalytic electroless deposition is schematically showed in Figure 13. Even in the case of electroless deposition, homogeneous dispersion of CNTs in the plating bath is important. The introduction of functional groups on the surface of CNTs likely effective to increase deposition sites, resulting in CNTs coated by metal films and not metal particles.

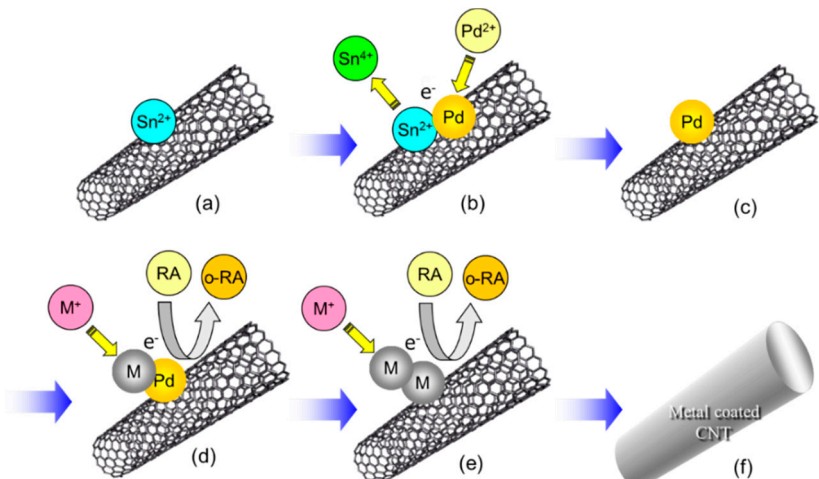

**Figure 13.** A fabrication process of metal-coated CNTs by electroless deposition. (**a**) adsorption of $Sn^{2+}$ on CNT, (**b**) Redox reaction: $Sn^{2+}$ to $Sn^{4+}$ and $Pd^{2+}$ to Pd, (**c**) Pd catalyst nanoparticle on CNT, (**d**) Redox reaction on the Pd catalyst particle: reducing agent (RA) to oxidized reducing agent (o-RA) and metal ion ($M^+$) to metal (M), (**e**) redox reaction in (**d**) continues even on the deposited M catalytically, (**f**) CNT is coated entirely with M, results in metal-coated CNT. The processes of (**a**–**c**) are pre-treatment processes and those of (**d**–**f**) are electroless deposition processes.

#### 3.2. Metal-Coated CNTs

Regarding the numbers of published articles on metal-coated CNTs by electroless deposition, those on Ni-P alloy-coated CNTs are large. In the following section, the summaries of articles on various metal-coated CNTs by electroless deposition are described. However, the articles on metal-particle deposited CNTs by electroless deposition for the use of catalyst are excluded. Fabrication conditions in these articles are listed in Table 3.

#### 3.2.1. Ni- or Ni Alloy-Coated CNTs

Li et al. [125] fabricated Ni-P alloy-coated MWCNTs by electroless deposition and evaluated their magnetic properties. They used standard processes, that is sensitization, activation, and electroless deposition, to fabricate Ni-P alloy coatings on MWCNTs. Ang et al. [126] proposed a single-step activation method which used a mixture of $Sn^{2+}$ and $Pd^{2+}$. Kong et al. [127] studied the conditions to form continuous Ni-P alloy layer on MWCNTs. Graphitized MWCNTs which have little functional groups on their surfaces have strong hydrophobicity, and consequently homogeneous dispersion of CNTs during the pre-treatment is difficult. Using a dispersing agent during the pre-treatment processes, graphitized MWCNTs were homogeneously coated with Ni-P alloy [128]. Wang et al. [129] used graphitized MWCNTs and fabricated Ni-P-coated MWCNTs employing a dispersant of MWCNTs during electrodeposition in addition to pre-treatment processes. Pure-Ni-coated and Ni-B alloy-coated MWCNTs were also fabricated [130,131] (Figure 14). Li et al. [132] prepared Ni-P alloy-coated SWCNTs and analyzed the microstructure of the Ni-P alloy layers. Jagannatham et al. [133] prepared Ni-P alloy-coated MWCNTs using an arc discharge synthesized MWCNTs. Ergul et al. [134] analyzed microstructures of Ni-P- and Co-P-coated MWCNTs. Poor wettability of CNTs with molten metals is a

serious problem to fabricate metal/CNT composites by metallurgical methods. Ni-P alloy-coated and Au/Ni-P-alloy double-coated MWCNTs were fabricated and their improved wettability with a molten Al was demonstrated [135]. Mani et al. [136] prepared Fe-50Co composites reinforced by Ni-P alloy coating and evaluated their magnetic and mechanical properties. Resin/CNT composites have been expected as next-generation lightweight electromagnetic interference (EMI) shielding materials [137,138]. Zhao et al. [139] prepared Ni-P-coated MWCNTs and formed epoxy/Ni-P-coated CNT composites and evaluated their EMI shielding properties. Park et al. [140] prepared Fe-B alloy/Ni-P alloy double-layer-coated MWCNTs and formed their epoxy resin composites and studied their EMI properties. Qi et al. [141] fabricated a Ni-P alloy/CNT composite plated cotton fabrics and demonstrated their superior EMI properties.

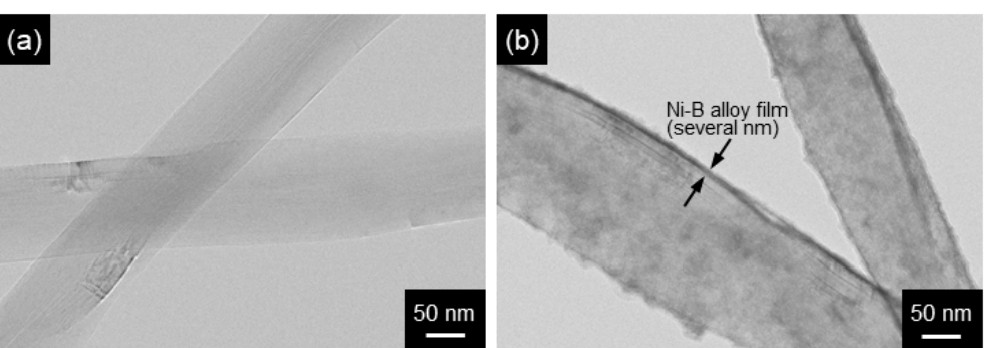

**Figure 14.** STEM images of Ni-B alloy-coated MWCNTs by electroless deposition: MWCNTs (**a**) before and (**b**) after electroless deposition. Ni-B alloy films of several nm thick are seen. (Figure 14 is adapted from reference [131]).

### 3.2.2. Other Metal-Coated CNTs

Chen et al. [142] prepared Co-P-coated MWCNTs using a sodium diphosphate as the reducing agent and evaluated their magnetic properties. Wang et al. [143] fabricated Cu-coated MWCNTs using a glyoxylic acid as the reducing agent. Daoush et al. [144] prepared Cu-coated MWCNTs and sintered them to form Cu/MWCNT composites. Feng et al. [145] prepared Ag-coated MWCNTs using a formaldehyde as the reducing agent. Mohammed et al. [146] fabricated Al-coated MWCNTs from an $AlCl_3$-urea ionic liquid using a lithium aluminum hydride as the reducing agent.

**Table 3.** Fabrication conditions of metal-coated CNTs by electroless deposition.

| Metal | CNT | Pre-Treatment of CNT | Reducing Agent | Surfactant | Remarks | Year | Ref. |
|---|---|---|---|---|---|---|---|
| Ni-P | MWCNT | $Sn^{2+}$ sensitization + $Pd^{2+}$ activation | $NaH_2PO_2$ | Non | Microstructure, Co-coated CNTs | 2020 | [134] |
| Ni-P | MWCNT | Introduction of -COOH on CNT + $Pd^{2+}$ | $NaH_2PO_2$ | Non | EMI properties, Cotton fabric substrate | 2020 | [141] |
| Ni-P | MWCNT | $Sn^{2+}$ sensitization + $Pd^{2+}$ activation | $NaH_2PO_2$ | Non | Arc discharge synthesized CNTs | 2015 | [133] |
| Ni-P | MWCNT | $Sn^{2+}$/$Pd^{2+}$ commercial product | $NaH_2PO_2$ | Non | Fe-50Co composites, magnetic properties | 2014 | [136] |
| Au/Ni-P | MWCNT | $Sn^{2+}$ sensitization + $Pd^{2+}$ activation | $NaH_2PO_2$ | Polyacrylic acid (Pre-treatment) | Improved wettability with molten Al | 2012 | [135] |
| Fe-B/Ni-P | MWCNT | $Sn^{2+}$ sensitization + $Pd^{2+}$ activation | $NaH_2PO_2$, $KBH_4$ | Non | Microwave absorbing properties | 2011 | [140] |
| Ni-P | SWCNT | $Sn^{2+}$ sensitization + $Pd^{2+}$ activation | $NaH_2PO_2$ | Non | Microstructure of Ni-layer | 2011 | [132] |
| Ni-B | MWCNT | $Sn^{2+}$ sensitization + $Pd^{2+}$ activation | $(CH_3)_2NH \cdot BH_3$ | Polyacrylic acid (Pre-treatment) | Graphitized MWCNTs Heat treatment | 2011 | [131] |

**Table 3.** *Cont.*

| Metal | CNT | Pre-Treatment of CNT | Reducing Agent | Surfactant | Remarks | Year | Ref. |
|---|---|---|---|---|---|---|---|
| Ni | MWCNT | $Sn^{2+}$ sensitization + $Pd^{2+}$ activation | $N_2H_4$ | Polyacrylic acid (Pre-treatment) | Graphitized MWCNTs Magnetic properties | 2010 | [130] |
| Ni-P | MWCNT | $K_2Cr_2O_7+H_2SO_4$ $Sn^{2+}$ sensitization + $Pd^{2+}$ activation | $NaH_2PO_2$ | Non | Microwave absorbing properties, Ni-N alloy | 2008 | [139] |
| Ni-P | MWCNT | $HNO_3$ $Sn^{2+}$ sensitization + $Pd^{2+}$ activation | $NaH_2PO_2$ | Diallyl-dimethylammonium chloride | Graphitized MWCNTs | 2005 | [129] |
| Ni-P | MWCNT | $Sn^{2+}$ sensitization + $Pd^{2+}$ activation | $NaH_2PO_2$ | Polyacrylic acid (Pre-treatment) | Graphitized MWCNTs | 2004 | [128] |
| Ni-P | MWCNT | $Sn^{2+}$ sensitization + $Pd^{2+}$ activation | $NaH_2PO_2$ | Non | Continuous Ni-layer | 2002 | [127] |
| Ni-P | MWCNT | Mixed $Pd^{2+}/Sn^{2+}$ | $NaH_2PO_2$ | Non | Pd-coated CNTs | 1999 | [126] |
| Ni-P | MWCNT | $Sn^{2+}$ sensitization + $Pd^{2+}$ activation | $NaH_2PO_2$ | Non | Magnetic property | 1997 | [125] |
| Al | MWCNT | $Sn^{2+}/Pd^{2+}$ commercial product | $LiAlH_4$ | Non | Non-aqueous bath: $AlCl_3$-urea | 2020 | [146] |
| Ag | MWCNT | $H_2SO_4 + HNO_3$ $Sn^{2+}$ sensitization + $Pd^{2+}$ activation | HCHO | Non | Interfacial adhesion of composites | 2004 | [145] |
| Cu | MWCNT | Sulphoric acid + $HNO_3$ $Sn^{2+}$ sensitization + $Cu^{2+}$ activation | HCHO | Non | Electrical and mechanical properties | 2009 | [144] |
| Cu | MWCNT | $HNO_3$ $Sn^{2+}$ sensitization + $Pd^{2+}$ activation $HNO_3$ | CHOCOOH | Diallyl-dimethylammonium chloride | Graphitized MWCNTs | 2004 | [143] |
| Co-P | MWCNT | $K_2Cr_2O_7+H_2SO_4$ $Sn^{2+}$ sensitization + $Pd^{2+}$ activation | $NaH_2PO_2$ | Non | Heat-treatment | 2000 | [142] |

## 4. Metal/CNT Composites by Electrodeposition Using CNT Templates (Sheet, Yarn)

CNT templates, such as CNT sheets [147–150] and CNT yarns or fibers [151–154], have been developed and their various practical applications have been researched. Although a single CNT has a high electrical conductivity, electrical conductivities of those templates are far less than metals such as Cu, due to the contact resistance between each CNT of which they consist. Therefore, metallization of the CNT templates is a promising process to give them enough electrical conductivity. On the contrary, CNTs have strong anisotropy in electrical and thermal properties [155]. Therefore, the orientation of CNTs which make up the templates is also important in order to achieve the expected properties of metal/CNT composites. In the following section, the summaries of articles on metal/CNT composites by electrodeposition using CNT templates are described. Fabrication conditions in these articles are listed in Table 4.

### 4.1. Metal/CNT Composite Using CNT Sheet (Paper, Film)

Jin et al. [156] and Shuai et al. [157] prepared laminar Cu/MWCNT composites reinforced by super-aligned CNT thin films and demonstrated their improved mechanical properties. The laminar composites were fabricated by repeating the Cu electrodeposition on the super-aligned CNT film and pasting the CNT sheet on the surface of deposited Cu. Hou et al. [158] prepared laminar Ni/MWCNT composites reinforced by the super-aligned CNT thin film in the same process and demonstrated their excellent mechanical properties. Shuai et al. [159] fabricated Cu/MWCNT composites reinforced by the Ni-coated super-aligned CNT shin film in the same way and showed improved mechanical and electrical properties. Tao et al. [160] prepared laminated Cu/CNT/Cu composites by electrodeposition employing MWCNT sheets, and studied their electrical properties. They used acid sulfuric Cu plating baths with and without additives (polyethylene glycol and chloride ions) and an alkaline Cu plating bath and compared the morphologies of the electrodeposited Cu on the CNT sheets. Cu/SWCNT composite sheets which contain

Cu throughout the interior were fabricated through one-step electrodeposition by the combination of additives [161] (Figure 15).

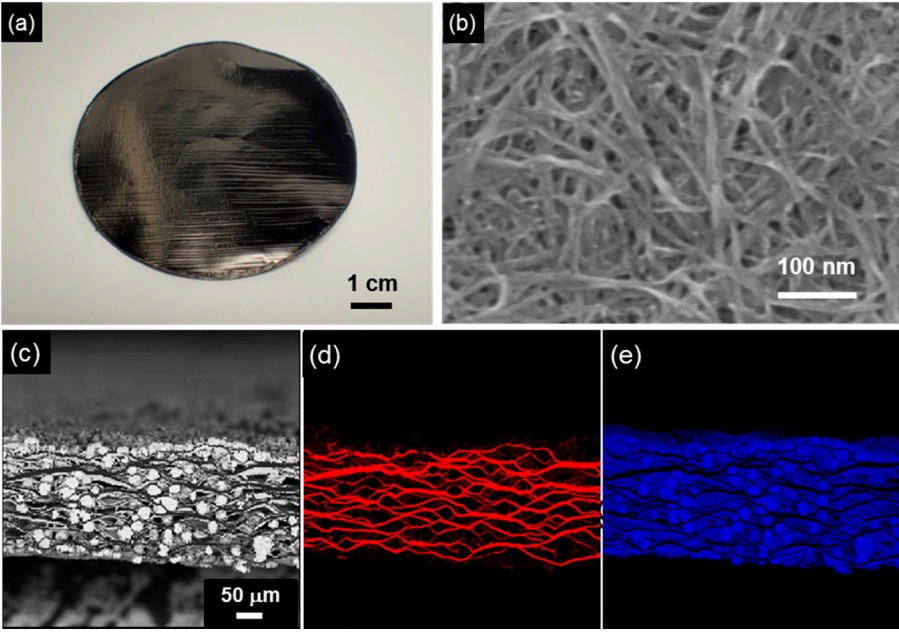

**Figure 15.** SWCNT/Cu composite sheet by electrodeposition using a SWCNT sheet as the template. (**a**) Appearance of SWCNT sheet, (**b**) Surface SEM image of the SWCNT sheet, (**c**) Cross-sectional SEM image of the CNT/Cu composite sheet by electrodeposition using a plating bath with additives, (**d**) C distribution of (**c**), and (**e**) Cu distribution of (**c**). (Figure 15 is adapted from [161]).

### 4.2. Metal/CNT Composite Using CNT Yarn (Fiber)

Lakshman et al. [162] prepared metal/MWCNT composite yarns by coating the MWCNT yarns with various metals. They proposed a new technique "self-fuelled electrodeposition". The resultant composite yarns show improved electrical conductivities and a reduction in tensile strength. Xu et al. [163] demonstrated a continuous process, which combines fiber spinning, CNT anodization, and Cu deposition, and showed improved tensile strengths of the resultant Cu/MWCNT yarns. Chen et al. [164] prepared Ag/CNT and Pt/CNT composite yarns using a potentiostatic and a cyclic voltammetry methods, respectively, and showed their improved tensile strengths and electrical conductivities. Hannula et al. [165] prepared Cu/MWCNT yarns and studied their electrical conductivities. They analyzed the cross-sections of the composite yarns and confirmed that the MWCNT yarns were filled with Cu. One of the outstanding properties of CNTs is their extremely high current carrying capacity (ampacity) [166–168]. Subramaniam et al. [169] showed that Cu/SWCNT composites which were fabricated by a two-step electrodeposition of aligned SWCNT forest possessed 100-fold higher current carrying capacity compared with Cu. The two-step electrodeposition consisted of electrodeposition using an organic bath and successive electrodeposition using an aqueous bath. Sundaram et al. [170–172] fabricated Cu/MWCNT composite yarns which were perfectly filled with Cu using the similar two-step process and demonstrated their superior electrical properties and solderability. Subramaniam et al. [173,174] and Chen et al. [175] applied this technology to other electronics applications. Rho et al. [176] prepared Cu/CNT composite fibers and demonstrated their superior current carrying capacity. By the combination of additives in plating bath, Cu/MWCNT composite yarns filled with Cu were fabricated one-step electrodeposition [177] (Figure 16). Park et al. [178] fabricated graphene-assisted CNT/Cu composite fibers and demonstrated their improved mechanical and electrical properties.

**Table 4.** Fabrication conditions of Metal/CNT Composites by Electrodeposition using CNT Template.

| CNT Template | Feature of CNT Template | Metal | Plating Bath | Remarks | Year | Ref. |
|---|---|---|---|---|---|---|
| MWCNT film | Super-aligned | Cu, Ni | Acid sulfuric bath + glucose Dull Watts Bath | Improved mechanical and electrical properties | 2019 | [159] |
| MWCNT film | Super-aligned | Ni | Dull Watts Bath | Improved mechanical properties | 2019 | [158] |
| SWCNT paper (Bucky paper) | Orientation: in-plane direction | Cu | Acid sulfate bath + polyethylene glycol + $Cl^-$ + bis(3-sulfopropyl) disulfide + Janus green B | One-step electrodeposition by a combination of additives | 2017 | [161] |
| MWCNT paper | Super-aligned | Cu | Acid sulfuric bath + glucose + polyethylene glycol + $Cl^-$ Alkaline bath (EDTA, Citrate) | Electrical conductivity | 2017 | [160] |
| MWCNT film | Super-aligned | Cu | Acid sulfuric bath + glucose | Improved mechanical properties | 2016 | [157] |
| MWCNT film | Super-aligned | Cu | Acid sulfuric bath + glucose | Improved mechanical properties | 2015 | [156] |
| SWCNT yarn | Straight | Cu | Acid sulfate bath | Graphen growth on the surface of electrodeposited Cu | 2021 | [178] |
| MWCNT yarn | Twisted | Cu | Acid sulfate bath + polyethylene glycol + $Cl^-$ + bis(3-sulfopropyl) disulfide + Janus green B | One-step electrodeposition by a combination of additives | 2020 | [177] |
| CNT yarn | Straight | Cu | Acid sulfate bath | Superior current carrying capacity | 2018 | [176] |
| MWCNT yarn | Twisted | Cu | $(CH_3COO)_2$ + $CH_3CN$ Acid sulfuric bath | Effect of CNT yarn density | 2018 | [172] |
| MWCNT yarn | Twisted | Cu | Cu $(CH_3COO)_2$ + $CH_3CN$ Acid sulfuric bath | Two-step electrodeposition Uniform composite wire | 2017 | [171] |
| MWCNT yarn | Twisted | Cu | $(CH_3COO)_2$ + $CH_3CN$ Acid sulfuric bath | Two-step electrodeposition Electrical properties, Solderability, | 2017 | [170] |
| MWCNT yarn | Straight | Cu | Acid sulfuric bath | Electrodeposition of Cu interior of CNT yarn | 2016 | [165] |
| MWCNT yarn | Twisted | Ag, Pt | $KNO_3$+$AgNO_3$ $H_2SO_4$ + $H_2Pt_6Cl_6$ | Improved tensile strength and electrical conductivity | 2013 | [164] |
| MWCNT yarn | Twisted | Cu | Acid sulfuric bath + octyl phenyl poly (ethylene gylcol) ether | Continuous process: fiber spinning, anodization, electrodeposition | 2011 | [163] |
| MWCNT yarn | Twisted | Au, Pd, Pt, Cu, Ag, Ni | Metal salt solution | Self-fueled electrodeposition Improved electrical conductivity | 2010 | [162] |

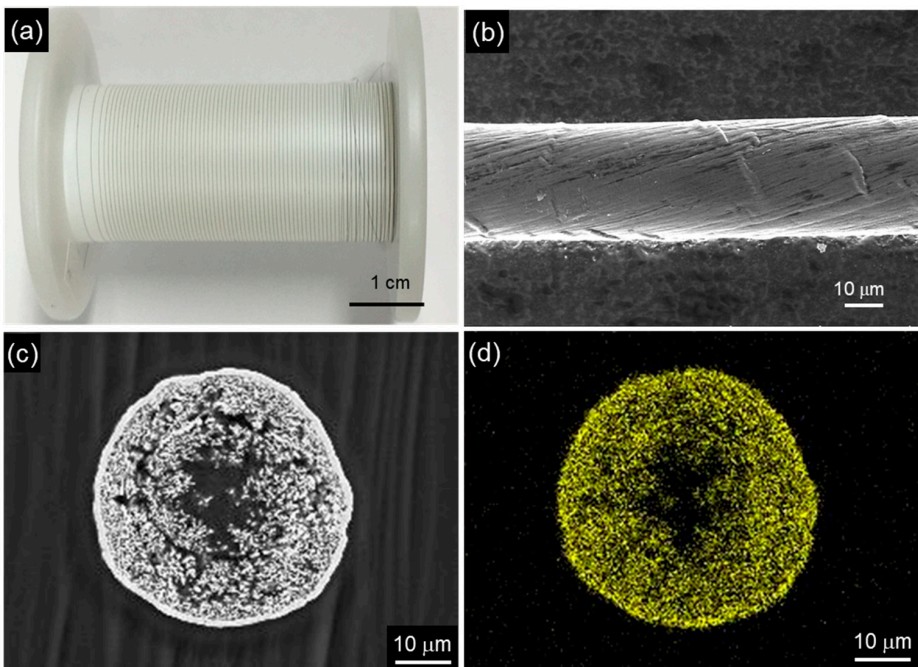

**Figure 16.** CNT/Cu composite yarn by electrodeposition using a CNT yarn as the template. (**a**) Appearance of CNT yarn, (**b**) SEM image of the CNT yarn, (**c**) Cross-sectional SEM image of CNT/Cu composite yarn by electrodeposition using a plating bath with additives, (**d**) Cu distribution of (**c**). (Figure 16 is adapted from [177]).

## 5. Future Challenges

### 5.1. Metal/CNT Composites by Composite Plating

Since numerous articles on excellent tribological properties and superior corrosion resistances of metal/CNT composites, especially Ni or Ni-alloy/MWCNT composites, have been published, commercial applications or commercial products using the composite plating are expected. The application for electrical contact materials is thought to be hopeful. Actually, some electrical contact parts using an Ag/CNT composite plating have been commercialized to some extent. The application for lithium-ion batteries, especially as current collectors of electrodes, can be expected soon enough. To increase the case study of practical applications of the composite plating, the elucidation of the mechanism on metal/CNT composite plating and the precise control of metal/CNT composites fabrication are essential. The evaluation of CNT content in metal/CNT composites prepared by composite plating is difficult and consequently accurate measuring methods should be proposed. Although the microstructures of the composites have been analyzed using TEM, etc., three-dimensional analysis of CNT contribution throughout the metal matrices should also be investigated.

### 5.2. Metal-Coated CNTs by Electroless Deposition

Metal-coated CNTs are used as row materials for powder metallurgy or filler materials for the composites such as resin composites. To enhance the magnetic shielding properties of resin composites, further studies on the metal-coated CNTs are expected for future work. Since metal-coated CNTs show the superior wettability with molten metals, fabrication of metal/CNT composites, such as Al/CNT composites, using melting method can be expected. The control of film thickness of deposited metal is important and the development of metal-coated CNTs with thinner film thickness are hopeful. Since this process is so called "powder plating" by electroless deposition, the mass production is difficult due to the rapid change in composition of plating baths, such as bath pH, especially in the case of CNTs which possess a large specific surface area. Therefore, the development of the system for the mass production of metal-coated CNTs are expected for future work.

### 5.3. Metal/CNT Composites Using CNT Template

The fabrication of metal/CNT composites using super-aligned CNT sheets makes sense for improving electrical and mechanical properties. Further studies are expected for future work. Cu/CNT composite yarns have been expected as lightweight electrical cables and, consequently, further research and developments are expected in future work. There has been progress made in the research and development of Cu/CNT composites, not only on the composite cables but also on other composites, such as electronic circuit wiring. The practical applications of these are expected in future work.

## 6. Conclusions

In this paper, a comprehensive review on the fabrication of metal/CNT composites by electrochemical deposition has been carried out. The number of articles on the fabrication of metal/CNT composite by electrochemical deposition has increased year by year. The fabrication process can be classified into three types: (1) composite plating by electrodeposition and electroless deposition, (2) metal coating on CNTs by electroless deposition, and (3) electrodeposition using CNT templates. In the composite plating, homogeneous dispersion of CNTs in plating baths is essential and, consequently, various processes, such as the addition of dispersants and introduction of hydrophilic groups on CNTs, have been studied. Numerous articles on Ni/CNT or Ni-P alloy/CNT composites by composite plating have been published and their excellent tribological properties and improved corrosion resistances have been reported. Many papers on Cu/CNT composites have also been published and their properties, such as electrical conductivity, have been investigated. The further elucidation of the mechanism of CNT composite plating process is expected. In the metal coating on CNTs by electroless deposition, the pre-treatments, such as sensitization and

activation, are important. Oxidization of CNTs is useful for coating CNTs perfectly. A lot of articles on Ni-P alloy-coated CNTs have been published. In the electrodeposition using CNT templates, many papers on Cu/CNT composites using CNT sheets and CNT yarns have been published and their electrical properties have been reported. The preparation process to deposit Cu not only on the surfaces but also on the interior of CNT templates is likely the key technical point.

The practical applications of these technologies are expected in future work.

**Funding:** This work was supported in part by the Japan Science and Technology Agency (JST), Adaptable and Seamless Technology Transfer Program through Target-driven R&D (A-STEP: Grant No. JPMJTM20E2).

**Institutional Review Board Statement:** Not applicable.

**Informed Consent Statement:** Not applicable.

**Data Availability Statement:** Data available in a publicly available repository.

**Conflicts of Interest:** The author declares no conflict of interest.

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
