# Peer review of "Fabrication of Metal/Carbon Nanotube Composites by Electrochemical Deposition"

_2673-3293, doi:10.3390/electrochem2040036_

Round 1

Reviewer 1 Report

Dear Author

Please find enclosed the review report to the manuscript submitted entitled "Fabrication of Metal/Carbon Nanotube Composites by Electrochemical Deposition".

The review entitled “Fabrication of Metal/Carbon Nanotube Composites by Electrochemical Deposition” is focused on the fabrication process of metal-CNTs composites through two electrochemical deposition methods: electrodeposition and electroless deposition.

This review presents a well-structured introduction, leading the reader to understand the main focus of this review. The paper is well written and carefully divided into subchapters for the readers to easily understand and to be easy to follow. The topic covered in the review is widely supported by the relevant expertise of the author in material processing and microstructural control engineering and structural/functional materials using electrochemical approaches.

I recommend this article for publication with minor suggestions addressed to works that might help to enrich this review for future publication:

  • In Line 291, the author refers the publication reported by Pereira et al., regarding the electrodeposition of Co/MWCNT composite films. In order to enrich the information provided in this paper, it would be important to address more research focused on non-aqueous electrolytes such as deep eutectic solvents (and ionic liquids) for the electrodeposition of metals on the MWCNTs matrix (which is demonstrated that it favours the carbon allotropes dispersion). Several papers have been published on the topic of metal-carbon composites using DES and ILs, so I think that this topic should be addressed. Below it is provided some papers that should be included in the main text:

Section 2.4.6 Au and Ag/CNT composites

  • Duc Chinh V, Speranza G, Migliaresi C, Van Chuc N, Minh Tan V, Phuong N-T. Synthesis of gold nanoparticles decorated with multiwalled carbon nanotubes (Au- MWCNTs) via cysteaminium chloride functionalization. SciRep 2019;9:5667. https://doi.org/10.1038/s41598-019-42055-7.

  • Characterization and electrochemical studies of MWCNTs decorated with Ag nanoparticles through pulse reversed current electrodeposition using a deep eutectic solvent for energy storage applications

https://doi.org/10.1016/j.jmrt.2021.08.031

  • Dobre N, P. _A A, Buda M, A._A. I L, ViS‚ an T. Electrochemical synthesis of silver nanoparticles in aqueous electrochemical synthesis of silver nanoparticles in aqueous electrolytes.

U.P.B Sci Bull 2014;76.

Section 2.4.2

The authors did not mention the work reported by Raja et al. focused on the electrodeposition of Cu–SWCNT Composites. The authors used a direct current electrodeposition and varied the CNT proportions without use surfactants or dispersing agents, or neither functionalizing the SWCNTs.

Raja PMV, Esquenazi GL, Gowenlock CE, Jones DR, Li J, Brinson B, Barron AR. Electrodeposition of Cu–SWCNT Composites. C. 2019; 5(3):38. https://doi.org/10.3390/c5030038

Other relevant papers

Chai, G.; Sun, Y.; Sun, J.J.; Chen, Q. Mechanical properties of carbon nanotube–copper nanocomposites. J. Micromech. Microeng. 2008, 18, 0350131–0350134. https://doi.org/10.1088/0960-1317/18/3/035013

Electroless plating

Daoush,W.M.; Lim, B.K.; Mo, C.B.; Nam, D.H.; Hong, S.H. Electrical and mechanical properties of carbon nanotube reinforced copper nanocomposites fabricated by electroless deposition process. Mater. Sci. Eng. A 2009, 513, 247–253.

Wright, K.D.; Gowenlock, C.E.; Bear, J.C.; Barron, A.R. Understanding the effect of functional groups on the seeded growth of copper on carbon nanotubes for optimizing electrical transmission. ACS Appl. Mater. Interfaces 2017, 9, 27202–27212.

2.4.8. Other metal/CNT Composites

Carbon–tin composite materials was reported by Brandão et al. in which it was incorporated oxidized multi-walled carbon nanotubes (ox-MWCNT) and pristine multi-walled carbon nanotubes (P-MWCNT) into a metallic tin matrix using electrodeposition in deep eutectic solvents composed by choline chloride and ethylene glycol.

Brandão, A.T.S.C.; Anicai, L.; Lazar, O.A.; Rosoiu, S.; Pantazi, A.; Costa, R.; Enachescu, M.; Pereira, C.M.; Silva, A.F. Electrodeposition of Sn and Sn Composites with Carbon Materials Using Choline Chloride-Based Ionic Liquids. Coatings 2019, 9, 798. https://doi.org/10.3390/coatings9120798

Typos

Page 2 Fig 1 Electrodeposition uisng CNT templates… please correct …to using…

Page 2 line 50 The electrochemical deposition is a nano-scall or atomic scall process to… Do the author pretend to say …a nano-scale or atomic scale process?

  • Table 1: Behabior à correct to “ behavior”

Page 8 line 287 … and demonstrated theie superior tribological…should be corrected to …and demonstrated their superior tribological…

Page 9 Line 318 .. from an AlCl3-1-ethyl-3-methylimidasolium chloride molten salt… should be corrected to … from an AlCl3-1-ethyl-3-methylimidazolium chloride molten salt

Page 16 Line 444 … fabricated Al coated MWCNTs from an Al3-urea ionic liquid using a lithium aluminum… the ionic liquid should be correct to … from an AlCl3-urea ionic liquid

Reviewer 2 Report

In this review, authors described the Fabrication, physicochemical, textural of Carbon nanotubes (CNT) with the electrochemical deposition synthesis method. The aforementioned review paper is well described. The overall discussion is impressive. Therefore, this manuscript has been accepted in its present form.

Reviewer 3 Report

Please see the attachment for detailed comments. Thank you.
